# P63 and P73 Activation in Cancers with p53 Mutation

**DOI:** 10.3390/biomedicines10071490

**Published:** 2022-06-23

**Authors:** Bi-He Cai, Yun-Chien Hsu, Fang-Yu Yeh, Yu-Rou Lin, Rui-Yu Lu, Si-Jie Yu, Jei-Fu Shaw, Ming-Han Wu, Yi-Zhen Tsai, Ying-Chen Lin, Zhi-Yu Bai, Yu-Chen Shih, Yi-Chiang Hsu, Ruo-Yu Liao, Wei-Hsin Kuo, Chao-Tien Hsu, Ching-Feng Lien, Chia-Chi Chen

**Affiliations:** 1School of Medicine, I-Shou University, No. 8, Yida Rd., Jiaosu Village Yanchao District, Kaohsiung 82445, Taiwan; candyhsu101@gmail.com (Y.-C.H.); lily920127@gmail.com (F.-Y.Y.); aaa0970179110@gmail.com (Y.-R.L.); danielwu0918@gmail.com (M.-H.W.); jenway74@isu.edu.tw (Y.-C.H.); 2Department of Medical Laboratory Science, I-Shou University, No. 8, Yida Rd., Jiaosu Village Yanchao District, Kaohsiung 82445, Taiwan; evafen88@gmail.com (R.-Y.L.); twruuq@gmail.com (S.-J.Y.); vivian08010917@gmail.com (Y.-Z.T.); droub9562@gmail.com (Y.-C.L.); baizhizhi2002@gmail.com (Z.-Y.B.); zoeyuliao@gmail.com (R.-Y.L.); 3Department of Biological Science and Technology, I-Shou University, No. 8, Yida Rd., Jiaosu Village Yanchao District, Kaohsiung 82445, Taiwan; shawjf@isu.edu.tw; 4Department of Otolaryngology-Head and Neck Surgery, E-DA Hospital, No. 1, Yida Rd., Jiaosu Village Yanchao District, Kaohsiung 82445, Taiwan; intuition430@gmail.com; 5Department of Physical Therapy, I-Shou University, No. 8, Yida Rd., Jiaosu Village Yanchao District, Kaohsiung 82445, Taiwan; sean060211@gmail.com; 6School of Chinese Medicine for Post-Baccalaureate, I-Shou University, No. 8, Yida Rd., Jiaosu Village Yanchao District, Kaohsiung 82445, Taiwan; ed103797@edah.org.tw; 7Department of Pathology, E-Da Hospital, I-Shou University, No. 1, Yida Rd., Jiaosu Village Yanchao District, Kaohsiung 82445, Taiwan

**Keywords:** p53, p63, p73, mutation, gain of function, aggregation, anti-cancer drugs

## Abstract

The members of the p53 family comprise p53, p63, and p73, and full-length isoforms of the p53 family have a tumor suppressor function. However, p53, but not p63 or p73, has a high mutation rate in cancers causing it to lose its tumor suppressor function. The top and second-most prevalent p53 mutations are missense and nonsense mutations, respectively. In this review, we discuss possible drug therapies for nonsense mutation and a missense mutation in p53. p63 and p73 activators may be able to replace mutant p53 and act as anti-cancer drugs. Herein, these p63 and p73 activators are summarized and how to improve these activator responses, particularly focusing on p53 gain-of-function mutants, is discussed.

## 1. Introduction to the p53 Family

The p53 family has three members, p53, p63, and p73 [1,2,3]. TA (transactivation) isoforms of p53 family members are tumor suppressor genes [4,5]. p53 has a high frequency of mutation in cancers causing loss of its tumor suppression function [6,7]; however, p63 and p73 are rarely mutated in cancers [8,9,10]. In this review, we briefly introduce each of the members with an emphasis on the most common mutations of p53 making it nonfunctional. Further, we summarize p63 and p73 activators that can replace them to obtain a similar tumor suppressor function in the p53 family. Some p53 mutants can obtain oncogenic function as a gain of function similar to an oncogene [11,12]. We also discuss how to improve p63 or p73 activator drug response in p53 gain-of-function mutation cancer cells.

### 1.1. p53

*p53* was the second tumor suppressor gene identified, although p53 was actually discovered in 1979, before the first tumor suppressor gene *Rb*, which was cloned in 1986 [13]. p53 has a function in apoptosis, cell cycle arrest, autophagy, metabolism, DNA repair, translational control, and feedback mechanisms [14,15]. p53 knockout mice are prone to the spontaneous variety of tumors by 6 months of age (~age 34 in humans) [16]. p53 has an average ~50% mutation rate in cancers [6,7,17]. According to the Catalogue of Somatic Mutations in Cancer (COSMIC) database [18], there are 177,561 unique clinical samples with 50,215 unique samples having p53 mutations (Figure 1A). The top p53 mutation type is missense mutation accounting for 62.74% of mutations, and the second most prevalent mutation of p53 is a nonsense mutation which accounts for 10.72% of mutations. Eight missense mutations R175H, G245S, R248Q, R248W, R249S, R273H, R273S, and R282W called hotspot mutations account for ~28% of all p53 mutations identified in cancers [19]. p53 germline mutation can cause Li-Fraumeni syndrome (LFS) which is a hereditary syndrome with a relatively early age of cancer diagnosis usually before the age of 36 [20]. This syndrome is characterized by the early onset of various types of cancer such as soft-tissue, breast, brain, leukemia, lymphoma, gastrointestinal, head and neck, kidney, larynx, lung, skin, ovary, pancreas, prostate, testis, thyroid, and adrenocortical cancers [21,22]. Some congenital p53 mutations of the LFS are similar to those acquired by p53 hotspot mutations in cancers such as G245S [23,24], but some congenital p53 mutations only appear in LFS as germ-line specific mutations such as R337H [25,26]. In this review, we focus on drug therapy of nonsense and missense mutations of p53. The missense mutation is divided into two types, loss-of-function mutations, and gain-of-function mutations. We focus particularly on p63 and p73 reactivation with drug treatment to replace part of the p53 tumor-suppressor function.

### 1.2. p63

*p63* was cloned in 1998 [27]. The major C-terminal p63 isoforms are p63α, while dominant-negative ΔNp63 was the predominant N-terminal isoform in most tissues from the p73-High/p63-High group [28]. p63 is a rare rarely mutated in cancers [8]. According to the COSMIC database [18], there are 55,869 unique clinical samples with 2160 unique samples having p63 mutations (Figure 1B). The nonsense mutation rate of p63 is only 2.97%, and the missense mutation rate of p63 is 32.34%. Although p63 is rarely mutated in cancers, distinct p63 germline mutation can cause several different types of abnormal development issues. Ectrodactyly, ectodermal dysplasia, and cleft lip/palate (EEC) syndrome are mainly characterized by severe ectrodactyly and limb defects with a p63 missense mutation in the middle of the DNA binding domain [29]. The distinguishing features of ankyloblepharon-ectodermal defects-cleft lip/palate (AEC) syndrome are ankyloblepharon, congenital erythroderma, skin fragility, atrophy, palmoplantar hyperkeratosis, and extensive skin erosions with p63 missense mutation in the C-terminal the sterile-α-motif (SAM) domain and transcriptional inhibitory (TI) domains [30,31]. Different p63 mutations can also cause isolated split hand/foot malformation (SHFM) nonsyndromic diseases with a missense mutation of K193E and K194E and nonsense mutants of Q634X and E639X on TAp63α [32]. p63 knockout mice have been found to fail to form the stratified epidermis, limbs, teeth, mammary glands, and postnatal lethality due to dehydration [33,34].

### 1.3. p73

*p73* was cloned in 1997 [35]. Compared to p63, p73 also commonly expresses α isoforms [28] and the expression of TAp73 is higher than ∆Np73 [36]. p73 is rarely mutated in cancers [9,10]. According to the COSMIC database [18], there are 42,580 unique clinical samples with 879 unique samples with p73 mutations (Figure 1C). The nonsense mutation rate of p73 is only 2.28%, and the missense mutation rate of p73 is 27.33%. There are no reports about p73 germline mutation in relation to any type of genetic disorder or syndrome. But hydrocephalus, hippocampal dysgenesis, and pheromonal defects can be found in p73 knockout mice [37].

## 2. Types of p53 Mutations in Cancers

### 2.1. Nonsense Mutation

p53 nonsense mutations comprise ~10% of all p53 mutants (Figure 1A); the actual p53 nonsense mutation rate is higher than the average ~5% [38]. There are three pre-stop DNA codons, TAA, TAG, and TGA. Nonsense mutation leads to the generation of premature termination codons (PTC), which leads to nonsense-mediated mRNA decay (NMD), resulting in the inability to express full-length proteins and extremely low expression levels of truncated proteins [39]. Two mechanisms are known to regenerate full-length proteins, one is to inhibit NMD, and the other is for PTC readthrough. It is known that aminoglycoside drugs, such as G418 and gentamicin, can inhibit NMD and promote p53 PTC readthrough [40,41]. But these drugs are highly toxic and cannot be used in clinical practice. 2,6-Diaminopurine (DAP), can inhibit the activity of putative ribosomal RNA methyltransferase 1 (FTSJ1) to increase the capacity of tRNATrp to recognize the UGA stop codon to promote p53 PTC readthrough, but this drug does not have the ability to inhibit NMD [42]. Furthermore, DAP is only effective for nonsense mutations of TGA but not TAA or TAG [42], and this greatly reduces the available targets. In addition, some phthalimide derivatives and antimalarial drug quinines can promote the p53 PTC read-through ability of G418 to increase the proportion of full-length p53 and to reduce the expression of the truncated protein, but these drugs alone have no effect on PTC read-through [43,44]. Non-aminoglycoside drugs, such as Ataluren (PTC124), also increase the read-through ability of PTC without the ability to inhibit NMD, and PTC124 has been used in clinical phase II or III trials to treat genetic diseases with specific nonsense mutations [45,46]. PTC124 can also promote p53 PTC readthrough [47]. A recent study has shown that CC-885 and CC-90009 can inhibit NMD; of note, the effective concentration of CC-885 for treatment of p53 nonsense mutation with TAA is only one-tenth of that of CC-90009 [48].

### 2.2. Loss-of-Function Mutants

p53 missense mutations contain both loss of function and gain of function. p53 mutation is mainly located at the N-terminal transactivation domain or middle DNA binding domain [2]. In addition, several point mutants still have normal DNA binding function [49,50,51,52]; most p53 mutations within the N-terminal transactivation domain or the DNA binding domain lose their transactivation function or DNA binding function causing loss of their tumor suppressor functions such as cell cycle checkpoint controls and apoptosis [53,54,55]. These p53 mutations can associate with p63 and p73, whereas wild-type p53 cannot [56,57]. Therefore, loss-of-function p53 mutants act in the same way as the ∆N isofroms of the p53 family having a dominant-negative effect to repress the functions of normal TA isoforms of p53 family members [57,58].

### 2.3. Gain-of-Function Mutations

Some p53 mutants can obtain some oncogenic functions such as cell migration, invasion, and metastasis to enhance tumorigenesis [59], and these p53 mutants are called gain-of-function mutants. The acquisition of p53 gain-of-function is via three mechanisms [60,61]. First, mutant p53 can directly bind to the novel binding site with a p53 non-canonical sequence to activate several oncogenic genes [51]. Second, mutant p53 can act as a co-activator to bind to other transcription factors to activate some oncogenic genes [62]. Third, mutant p53 can bind to other tumor suppressive-type transcription factors to cause loss of transcription ability [63]. Some p53 mutants can become aggregated in several types of cancer, such as breast, lung ovary, colorectal, and head and neck cancers [64,65,66,67]. It is known that these aggregations of mutant p53 can sequester other tumor suppressor genes as a third mechanism to cause p53 gain of function [68,69,70].

## 3. Factors Influencing p53 Mutant Gain of Function

Several factors have been reported to influence p53 mutant gain of function (Figure 2A). The heat shock protein 70 (HSP70) has been reported to enhance mutant p53 aggregation [71], and heat shock protein 90 (HSP90) can repress mutant p53 aggregation [71,72]. SIRT1 is an NAD+ dependent histone deacetylase that has been reported to deacetylate HSF1 to enhance HSF1 transcriptional activity to increase HSP70 expression [73]. NAMPT can enhance SIRT1 activity by increasing the amount of NAD+ [74]. p53 gain-of-function mutant can induce MYC [62,75], and MYC can enhance NAPMT [76]. Wild-type p53 can induce 14-3-3σ expression [77,78], and 14-3-3σ can promote MYC poly-ubiquitination and degradation [79]. Another article also reported that wild-type p53 can bind to G-quadruplexes on MYC promoter to repress MYC expression [80]. Both wild-type and non-gain-of-function p53 mutants can activate lincRNA-p21 through binding to G-quadruplexes on lincRNA-p21 promoter [81], and lincRNA-p21 can repress STAT3 [82]. STAT3 can bind to the HSP70 promoter to active HSP70 expression [83,84]. The relationships between factors that influence p53 mutant gain of function are summarized in Figure 2A.

MircoRNA and long-noncoding RNA are also key regulators of mutant p53. miR-150 can repress p53 [85,86,87]. An oncogenic type long-noncoding RNA LINC00460 can act as a sponge to repress miR-150 targeting to enhance mutant p53 expression [88]. Another tumor-suppressor type transcription factor, AP2, frequently interacts synergistically with p53 to activate downstream genes such as p21, CD82, and NEU4 [89,90,91,92]. AP2 can repress another oncogenic-type long-noncoding RNA, LINC00511 expression [93]. LINC00511 also can act as a sponge to repress miR-150 targeting [94]. So AP2 may also repress mutant p53 expression through the LINC00511-miR-150 axis (Figure 2B). AP2 has also been reported to decrease the amount of p53 [95].

## 4. p63 Activation Drugs

Because TAp63 isoforms have limited expression in organs [28], only a few reports have looked at TAp63 activation for anti-cancer purposes; they are summarized in Table 1. Bliotoxin was able to upregulate the levels of DAPK1 to induce TAp63 but not p53 or TAp73 expression to induce apoptosis in paclitaxel pretreated paclitaxel-resistant CaOV-3 and SK-OV-3 ovarian cancer cells [96]. Lovastatin was able to induce TP63 transactivation through phosphorylation of the AMPK-p38MAPK-TAp63 cascade to cause hypopharyngeal carcinoma FaDu cell death [97]. TAp63 can active PUMA (p53 upregulated modulator of apoptosis) [98,99], and interferon-α can induce TP63 and PUMA expression in hepatocyte derived cellular carcinoma cell line HuH7 cells [100].

miR-130b mimics can activate TAp63 and repress ΔNp63 to decrease cell viability in ovarian cancer Ovcar1-8 cells [101]. miR-124 mimics can activate TAp63 and repress ΔNp63 to inhibit cell growth in LoVo and SW480 colorectal cancer cells [102]. miR-140 can directly target p63 3′-UTR sequences to repress p63, and the miR-140 inhibitor can activate TAp63 in HGC-27 and BGC-823 gastric cancer cells to induce cell apoptosis [103].

**Table 1 biomedicines-10-01490-t001:** Table summarizing p63 activation drugs used to treat cancer cell lines. If there is no COSMIC ID or ATCC ID for a certain cell line from the p53 database [104], the PubMed ID is provided for information about the p53 status within indicated cell line. CDS, coding sequence. *, premature stop codons.

p63 Activation Drugs	Cell Line	COSMIC ID	ATCC ID	PubMed ID	Tissue	p53 Status	Zygosity	CDS Mutation	Protein Change	Ref.
gliotoxin	Caov-3	906825	HTB-75	-	Ovary	MUT	Homozygous	c.406C > T	p.Q136*	[96]
	SK-OV-3	905959	HTB-77	-	Ovary	MUT	Homozygous	c.267del	p.S90Pfs*33	[96]
Lovastatin	FaDu	906863	HTB-43	-	Head and neck	MUT	Heterozygous	c.743G > T	p.R248L	[97]
interferon-α	HuH7	907071	-	-	Liver	MUT	Homozygous	c.659A > G	p.Y220C	[100]
miR-130b mimics	Ovcar-8	905991	-	-	Ovary	MUT	Homozygous	c.376_396del	p.Y126_K132del	[101]
miR-214 mimics	LoVo	907790	-	-	Large intestine	Wild type	Homozygous	-	-	[102]
	SW480	-	CCL-228	-	Large intestine	MUT	Heterozygous	c.818G > A & 925C > T	p.R273H & P309S	[102]
miR-140 inhibitor	HGC-27	907055	-	-	Stomach	MUT	Heterozygous	c.455dup	p.P153Afs*28	[103]
	BGC-823	-	-	9999992	Stomach	wild type	Homozygous	-	-	[103]

## 5. p73 Activation Drugs

Over 20 research papers have covered TAp73 activation for anti-cancer purposes. This research is summarized in Table 2. RETRA was the first p73 activator identified. It was found to block the interaction between p73 and mutant p53 in A431 and SW480 p53 mutant cells to relieve p73 transactivation function [105]. NSC59984 can induce mutant p53 protein degradation and activate p73 in SW480 and DLD1 cells [106]. Prodigiosin can disrupt mutant p53 and p73 interaction [107], and it can not only activate TAp73 but can also repress the dominant-negative isoform ΔNp73 in SW480 and DLD1 cells [107,108]. Metformin can activate TAp73 through the LKB1-AMPK axis in HCT116 cells [109], and it also represses dominant-negative isoform ΔNp63 under low glucose conditions in FaDu, H596 and H292 cells [110]. Bortezomib can promote TAp73 activation to reduce colon cancer cell viability in both p53−/− HCT116 and HT29 cells [111]. Protoporphyrin can disrupt p73/MDM2 complexes to restore p73 transcriptional activity in p53−/− HCT116 cells [112]. Nutlin 3 is an MDM2 antagonist [113], and it also disrupts p73-MDM2 binding to enhance p73 function in IEC-6 and Caco2 cells [114]. Diallyl disulfide can upregulate TAp73 and downregulate ΔNp73 expression in HeLa cells to enhance carbon ion beam–induced apoptosis [115]. Abrus agglutinin can inhibit p73 and Snail interaction to cause p73 activation in FaDu cells [116]. Cinobufagin can repress AURKA serine/threonine kinase to decrease phosphorylation levels of p53 (S315 and S392) [117], but it can promote phosphorylation levels of p73 (Y99) to induce the apoptosis in Huh-7 cells [117]. Panobinostat caused p73 and p21 upregulation in TP53−/− Saos-2 osteosarcoma cells [118]. Thymoquinone can induce p73 expression to cause cell apoptosis in a p53-deficient acute lymphoblastic leukemia (ALL) Jurkat cell line [119,120]. Thymoquinone can inhibit Itch, the E3-ubiquitin ligase of p73 leading to the upregulation of tumor suppressor p73 in Jurkat cells, MDA-MB-468 cells, and HL60 cells [121,122]. Extracts of Piper betle leaf (PBL) can promote p73 expression to induce cell cycle arrest in Hep3B cells [123]. MEK1 inhibitor (PD98059 or PD184352) can induce TAp73 and reduce dominant-negative ΔNp73 in NB4 and K562 cell lines [124]. Etoposide can induce p73 expression in HOC313 and Ca9-22 cells [125], and it can also induce p73 expression in both HCT116 p53−/− and H1299 p53−/− cells [126].

miRNA-1180 can directly target p73 3′-UTR sequences to repress p73, and miR-1180 inhibitor can induce p73 in SK-NEP-1 cells [127]. miRNA-193a-5p can directly target p73 3′-UTR sequences to repress p73 [128], miRNA-193a-5p inhibitor can induce p73 in JHU-029 and MG63 cells [128,129]. miR-647 can directly target p73 3′-UTR sequences to repress p73, and miR-647 inhibitor can induce p73 in MGC-803 cell lines [130]. miR-323 can repress the expression of p73 in PC-3 cell lines, and miR-323 inhibitor can arrest the PC-3 cell cycle and cause apoptosis with increased expression of p73 downstream gene p21 [131]. miR106b mimic can directly target Itch, the E3-ubiquitin ligase of p73, and miR106b mimic can repress Itch to accumulate p73 in K562 cells [132].

## 6. Influence of Interactions between p53 Family Members on p63 or p73 Activators and the Importance of Combination Treatment Strategy

Several factors can influence p63 or p73 activation in p53 mutant cells. First, a high level of dominant-negative ∆Np63 or ∆Np73 expression in cells can block the TAp63 or TAp73 activity [57,58]. Second, gain-of-function mutant p53 can become aggregated and co-aggregate with TAp63 or TAp73 [68,133]. miR-130b and miR-124 mimics not only induce TA isoform p63 in p53 mutant cells, but also repress the relative proportion of ΔN isoform p63 to resolve the first issue [101,102]. Prodigiosin, metformin, diallyl disulfide, and MEK1 inhibitors not only induce TA isoform p73 in p53 mutant cells, but also repress the relative proportion of ΔN isoform p63 or p73 [107,110,115,124,134]. Recently we found that p73 activators RETEA and NSC59984 have a poor response in aggregative p53 mutant HNSCC cells compared to non-aggregative p53 mutant cells. Furthermore, using an NAMPT inhibitor to block p53 aggregation can enhance the anti-cancer effect of p73 activator RETEA and NSC59984 in p53 gain-of-function mutant HNSCC cells [135]. This combination treatment strategy comprised of a p73 activator and a p53 aggregation inhibitor was able to resolve the second issue to active p73 in p53 gain-of-function mutants. Because there are fewer p63 activators than p73 activators (Table 1 and Table 2), whether this combination treatment strategy with a p63 activator and p53 aggregation inhibitor is workable or not in p53 gain-of-function mutants still needs further investigation. However, it is likely that co-treatment with a p53 aggregation inhibitor can improve p63 and p73 drug response in p53 gain-of-function mutants.

## 7. Viral Proteins and the p53 Family

Some cells contain wild-type p53, but several of these cell lines may be infected by certain viruses to express viral protein(s) that interact with p53 family members to cause their loss of function. Hela cells were infected by oncogenic HPV type 16 (HPV16) with E6 oncoproteins to induce ubiquitin-dependent proteolytic degradation of wild-type p53 [136,137]. HPV18 E6 can interact with TAp63β but not the other p63 isoforms to induce degradation of the TAp63β [138]. HPV18 E6 can also interact with TAp73α or TAp73β to reduce its transcriptional activity but has no influence on p73 stability. Diallyl disulfide can still induce p73 to mediate the cell apoptotic program in Hela cells [115]. Other oncogenic viral proteins like the LANA protein of the Kaposi sarcoma virus, the BZLF1 protein of the Epstein-Barr virus, and the papain-like protease of the nonstructural protein 3 of SARS-CoV also can cause p53 degradation [139,140,141]. In addition, the HbX protein of the Hepatitis virus B and the NS2 proteins of hepatitis virus C can interact with p53 to reduce its transcriptional activity [142,143]. But whether all these oncogenic viral proteins can influence p63 and p73 or not still remains to be addressed. This means that p63 and p73 activators may also not be such powerful anti-cancer agents in p53 mutant cells infected by certain viruses. Interestingly, HPV18 E7 can inhibit the interaction of p53 and MDM2 to stabilize p53 to increase the transcriptional activation function of p53 [144]. The NS5 proteins of the Zika virus can interact with p53 to prolong the half-life of p53 to cause cell death [145]. Therefore, some viral protein(s) can also promote wild-type p53 function. Whether p63 and p73 can be activated by these viral protein(s) or not is also unknown.

## 8. p53 Isoforms in p53 Mutant Cancer Cells

p53, like p63 and p73, has p53 isoforms [146,147]. Two dominate-negative isoforms ∆133p53 and ∆160p53 are frequently expressed in p53 mutant cancer cells [148,149]. Wild-type p53 does not interact with p73 [56,57], but Δ133p53α, Δ133p53β, and Δ133p53γ isoforms can interact with p73 to alter p73 downstream genes RAD51, RAD52, and LIG4 [150]. One study reported that treatment of 5-lipooxygenase (5-LOX) peptide inhibitor, YWCS, can induce p73, Δ133p53, and Δ160p53 expression to prevent neurotoxicity in pre-treatment of Aβ_25–35_ in SH-SY5Y cells [151], so Δ133p53 and Δ160p53 play a neuroprotective role to offset the cell apoptotic role of p73. Wild-type p53 does not interact with p63 [56,57], but the Δ133p53α isoform can interact with p63 to repress the anti-proliferative activities of TAp63β [152]. Δ133p53β, but not Δ133p53α or Δ160p53β, can aggregate to deactivate its dominate-negative effect [153]. If free-form Δ133p53β is released from aggregates, Δ133p53β can interact with TAp63 or ∆Np63 to repress transactivation or to enhance an oncogenic effect separately [153]. The p53 isoforms likely also influence p63 and p73 activator response in p53 mutant cells; this it also needs to be clarified.

## 9. Discussion and Concluding Remarks

There are also several p53 activators that can directly bind to mutant p53 to convert it into a wild-type like structure [154,155,156]. These drugs directly reactivate and change p53 activity. Such drugs were not emphasized in this review. For example, PRIMA-1 and its methylated form PRIMA-1Met (APR-246) can directly bind to mutant p53 to refold it as a wild-type p53-like conformation [157,158,159], because mutant p53 can associate with p63 and p73, whereas wild-type p53 cannot [56,57]. Therefore, PRIMA-1Met has also been reported to activate p63 and p73 through relieving them from mutant p53 that had been converted into wild-type like structure [160,161,162].

In conclusion, this review focused on summarizing the drugs that can activate p63 or p73 in p53 mutant cancer cells. Besides synthetic chemicals and purified natural products, synthesized small biomolecules, like microRNA mimics or inhibitors have also been used to activate p63 and p73 in p53 mutant cancer cells (Table 1 and Table 2). It has also been pointed out in this review that viral proteins and p53 isoforms may also influence the drug response of p63 and p73 activators in p53 mutant cancers.

## Figures and Tables

**Figure 1 biomedicines-10-01490-f001:**
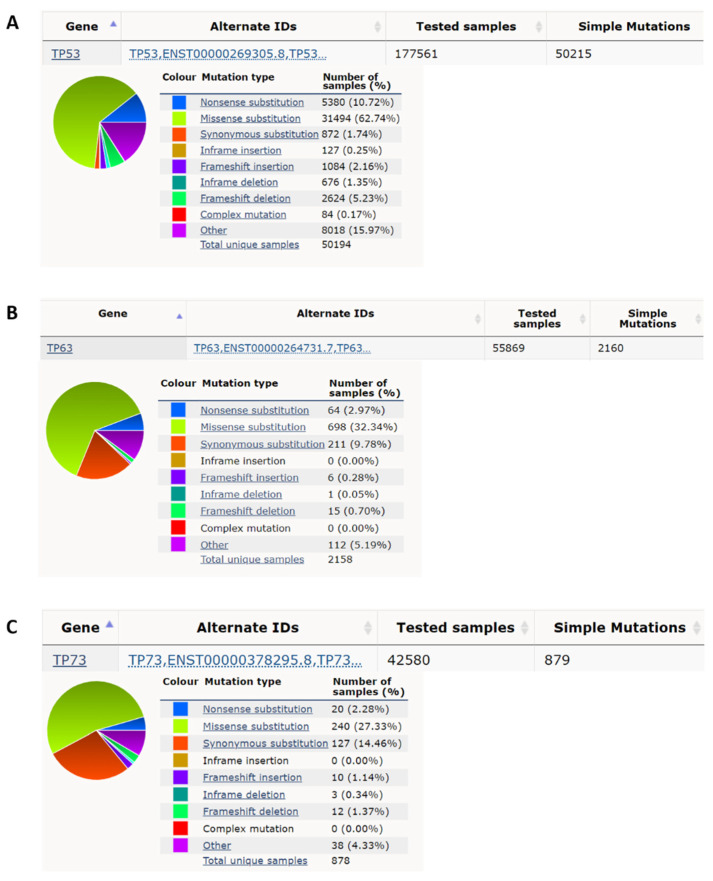
*p53* has a high nonsense and missense mutation rate compared to *p63* and *p73* in cancer. (**A**) *p53* has, respectively, a 10.72% and 62.74% nonsense mutation rate and missense mutation rate in all cancer mutation samples from the COSMIC database (https://cancer.sanger.ac.uk/cosmic; accessed date (29 April 2022) [18]. (**B**) *p63* has, respectively, 2.97% and 32.34% nonsense mutation rate and missense mutation rate in all cancer mutation samples. (**C**) *p73* has, respectively, 2.28% and 27.33%, nonsense mutation rate and missense mutation rate in all cancer mutation samples.

**Figure 2 biomedicines-10-01490-f002:**
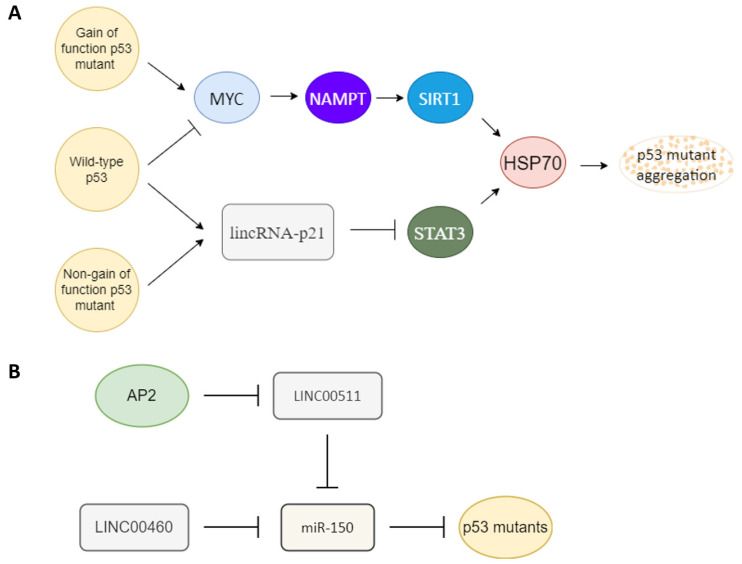
Key upstream molecules that influence p53 gain of function. (**A**) MYC and lincRNA-p21 are likely the key switch molecules that determine the gain-of-function or non-gain-of-function p53 signal to regulate the HSP70 expression to influence p53 mutant aggregation. Gain-of-function p53 mutant may induce the MYC-NAMPT- SIRT1-HSP70 axis to induce p53 auto-aggregation, and non-gain of function p53 mutants still can maintain non-aggregated p53 through the lincRNA-p21-STAT3-HSP70 axis. Wild-type p53 likely keeps its non-aggregated status through repression of MYC and upregulation of lincRNA-p21. (**B**) Oncogenic-type long-noncoding RNA LINC00511 and LINC00460 can act as a sponge to repress miR-150 to promote mutant p53 accumulation. A tumor-suppressor type transcription factor, AP2, can repress LINC00511 expression to decrease the amount of mutant p53.

**Table 2 biomedicines-10-01490-t002:** Table summarizing p73 activation drugs used to treat cancer cell lines. If there is no COSMIC ID or ATCC ID of a certain cell line from the p53 database [104], the PubMed ID is provided for information about the p53 status within indicated cell line. CDS, coding sequence. *, premature stop codons.

p73 Activation Drugs	Cell Line	COSMIC ID	ATCC ID	PubMed ID	Tissue	p53 Status	Zygosity	CDS Mutation	Protein Change	Ref.
RETRA	A431	910925	CRL-1555	-	Skin	MUT	Homozygous	c.818G > A	p.R273H	[105]
	SW480	-	CCL-228	-	Large intestine	MUT	Heterozygous	c.818G > A & 925C > T	p.R273H & P309S	[105]
NSC59984	SW480	-	CCL-228	-	Large intestine	MUT	Heterozygous	c.818G > A & 925C > T	p.R273H & P309S	[106]
	DLD1	-	CCL-221	-	Large intestine	MUT	Heterozygous	c.722C > T	p.S241F	[106]
Prodigiosin	SW480	-	CCL-228	-	Large intestine	MUT	Heterozygous	c.818G > A & c.925C > T	p.R273H &p.P309S	[107,108]
	DLD1	-	CCL-221	-	Large intestine	MUT	Heterozygous	c.722C > T	p.S241F	[107,108]
Metformin	HCT-116	905936	CCL-247	-	Large intestine	wild type	Homozygous	-	-	[109]
	FaDu	906863	HTB-43	-	Head and neck	MUT	Heterozygous	c.743G > T	p.R248L	[110]
	H596	908459	HTB-178	-	Lung	MUT	Homozygous	c.733G > T	p.G245C	[110]
	H292	753604	-	-	Lung	wild type	Heterozygous	-	-	[110]
Bortezomib	HT29	905939	HTB-38	-	Colon	MUT	Homozygous	c.818G > A	p.R273H	[111]
Nutlin 3	IEC-6	-	-	25230151	Small intestine; Epithelium	wild type	Homozygous	-	-	[114]
	Caco2	-	-	16418264	Large intestine	MUT	Homozygous	c.610G > T	p.E204*	[114]
Diallyl disulfide	HeLa	1298134	-	-	Cervix	wild type	Homozygous	-	-	[115]
Abrus agglutinin	FaDu	906863	HTB-43	-	Head and neck	MUT	Heterozygous	c.743G > T	p.R248L	[116]
Cinobufagin	Huh-7	907071		-	Liver	MUT	Homozygous	c.659A > G	p.Y220C	[117]
Thymoquinone	MDA-MB-468	908123	HTB-132	-	Breast	MUT	Homozygous	c.818G > A	p.R273H	[121,122]
	HL60	905938	CCL-240	-	Haematopoietic and lymphoid	NULL	Homozygous	c.1_1182del	No expression	[121,122]
Extracts of Piper betle leaf (PBL)	Hep3B	-	HB-8064	-	Liver	NULL	Homozygous	gross deletion	No expression	[123]
MEK1 inhibitor	NB-4	1323913	-	-	Haematopoietic and lymphoid	MUT	Homozygous	c.743G > A	p.R248Q	[124]
	K562	905940	CCL-243	-	Haematopoietic and lymphoid	MUT	Homozygous	c.406dup	p.Q136Pfs*13	[124]
Etoposide	HOC313	-	-	1570156	Mouth	MUT	Unknown	c.853G > A	p.E285K	[125]
	Ca9-22	753538	-	-	Mouth	MUT	Homozygous	c.742C > T	p.R248W	[125]
miR-1180 inhibitor	SK-NEP-1	909730	HTB-48	-	Kidney	MUT	Homozygous	c.733G > A	p.G245S	[127]
miRNA-193a-5p inhibitor	JHU-029	1298156	-	-	Head and neck	MUT	Heterozygous	c.323del	p.G108Vfs*15	[128,129]
	MG63	908131	-	-	Bone	wild type	Homozygous	-	-	[128,129]
miR-647 inhibitor	MGC-803	-	-	9999992	Stomach	wild type	Homozygous	-	-	[130]
miR-323 inhibitor	PC-3	905934	CRL-1435	-	Prostate	MUT	Homozygous	c.414del	p.K139Rfs*31	[131]
miR106b mimic	K562	905940	CCL-243	-	Haematopoietic and lymphoid	MUT	Homozygous	c.406dup	p.Q136Pfs*13	[132]

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
