# Peer review of "P63 and P73 Activation in Cancers with p53 Mutation"

_biomedicines, 2022, doi:10.3390/biomedicines10071490_

Round 1

Reviewer 1 Report

Manuscript ID: biomedicines-1760948
Type of manuscript: review, submitted to section: Molecular and
Translational Medicine,

Title: P63 and P73 Activation in P53 Mutant Cancers
Authors: Bi-He Cai *, Yun-Chien Hsu, Fang-Yu Yeh, Yu-Rou Lin, Rui-Yu Lu,
Si-Jie Yu, Jei-Fu Shaw, Ming-Han Wu, Yi-Zhen Tsai, Ying-Chen Lin, Zhi-Yu Bai,
Yu-Chen Shih, Yi-Chiang Hsu, Ruo-Yu Liao, Wei-Hsin Kuo, Chao-Tien Hsu,
Ching-Feng Lien *, Chia-Chi Chen *

In this review, the authors describe p63 and p73 activators that can replace non-functioning p53 to obtain a similar tumor suppressor function in the p53 family.

Overall, the manuscript is clear and well written. However, it contains some typing errors/language inconsistences which should be corrected:

for example, lines: 65, 135, 140, 144, 170, 253, 259, 287.

Figure 1 – the legend should be simplified – it contains the percentage data described already in the main text.

The gene names should be in italics.

Reviewer 2 Report

Dear authors, 

Thank you for an interesting article which addresses a very important topic with lots of future potential benefit.

My major concern is the lack of clarity around the members of the p53 family of isoforms and their interaction and overlapping functions. P53 family encompasess many isoforms of the p53, p73 and p63 and this should be incorporated in the introduction and in each of the section of the genes. It will help with the understanding of the text. Please check these review articles for further clarification and also some great papers you have already referenced. https://www.mdpi.com/1422-0067/20/24/6257,  https://www.mdpi.com/2072-6694/13/12/2885.

Also, the term "p53 mutant cancer" is rather not usual (in the title). Please revise if possible.

Furthermore, a sentence of intro and conclusion for each chapter is missing and would be beneficial for the understanding of the text flow. 

Some specific concerns:

line 22: "all have a tumor suppressor function" this is not correct, since certain isoforms have indeed oncogenic tendencies. Please revise.

line 25: please add "possible" or similar to "drug therapies". As far as I know, no approved therapy exists today.

line 26: i cant seem to understand causative relationship between "wide expression of p73" and the number of activators "there are many more p73 activators.

also line 25: Please define a little more what kind of drug therapies "In this review, we discuss drug therapies of nonsense mutation and missense mutation in p53." for example: "In this review, we discuss drug therapies of nonsense mutation and missense mutation in p53 which target p63 or p73 to reactivate p53 pathway".

Line 28: please remove "these" and define what is "these drug responses".

Line 33: as said above, the p53 family is not as simple as stated consistign of only three members. Please include information about isoforms and correct this sentence "All of the p53 family members are tumor suppressor genes"- it is not correct that all are supressors.

Line 36:  "In this review, we summarize p63 and p73 activators 36 that can replace non-functioning p53 to obtain a similar tumor suppressor function in 37 the p53 family." -here i dont see info about first part of the article: mutations, so please add this

eg: "In this review, we briefly introduce each of the members with the emphasis on the most common mutations of p53 making it nonfuctional. Further we  summarize p63 and p73 activators that can replace it to obtain a similar tumor suppressor function in the p53 family." or similar.

Line 39: unclear sentence: "We also discuss how to improve p63 or p73 activator  drug response in cancer cells with p53 gain-of-function mutants."

Line 45: correct "functions in" to "has a function in" or simiral

46: unclear: “feedback mechanisms“?

48: Why is the number “177,561 unique clinical samples with 50,215” important here. Please discuss. What is the total number of samples for example?

49: maybe change: “the top mutation” to “the top mutation type”

53: unclear “each contains”

54: What is the purpose of this part? “175, 179, 195, 196, 213, 217, 225, 229, 54
248, 249, 254, 273, 278, 281, and 298 [19].”

55: syndrome is written with small letter

60, 61, 62: I think “mutants” should be “mutations”. Also check other parts of the text for the same error.

63: “p53 knockout mice are prone to the spontaneous variety of tumours by 6 months of age (~age 34 in humans) [27]” maybe transfer to line 46 before talking about mutations?

67: I think the sentence should end with mutants instead of  “; here”. Change mutants to “mutations”.  

67: unclear “reactivity to drug treatment”

70:  Please check and correct “native” to “negative”

71: Please change: “is a rare” to “is rarely”. Same for line 75.

I feel that chapter 1.2  p63 is missing crucial information about why one would activate p63 in p53 mutant cancers.

Line 90: “but” should be “with â–³Np73α and TAp73α generally co-expressed in organs” or similar.

91: unclear: “So more p73 activators develop compared to p63 activators owing to the limited expression 91

of TAp63α.”

92: please change: “is rare mutation” to “is rarely mutated”

I feel that chapter 1.3 p73 is missing crucial information about why one would activate p73 in p53 mutant cancers

108: what is “theroretical rate“?

Figure would be great for chapter 2.1.

135: Please change: “N-termal“ to “N-terminal”

Line 139: Are you sure wt p53 can not be associated with p63 or p73?

140:  Please change “native” to “negative”

Chapter 2 seems unorganized. It is rather strange that 2.2. is not about missense mutations per se, but the title is loss of function? and 2.3. gain of function? 

153: is the aggregation “third” or “fourth mechanism”?

153: “to induce p53 gain of function” may be incorrect. You are talking about mechanism that was cause by the gain of function. Please revise.

Line 155. Please refer to Fig 2.

Fig 2 title: line 180: “cause” is incorrect. Please revise and change to “influence” or similar? There are some proteins missing like HSP90 from the figure.

Chapter 3 is missing conclusions. What was the purpose of this chapter and what are the main conclusions and future scopes? To target the listed proteins?

Line 191: unclear “Because TAp63 isoforms have limited expression in organs [29], only a few reports have looked at TAp63 activation for anti-cancer proposes”. Why would limited expression of isoform in organs would cause few reports about TAp63 activation for anti-cancer proposes?

Why would one even want to activate p63 for cancer research?

What are the main conclusions for chapter p63 activation?

Table 1: Define CDS. Maybe use wt for wild type p53 in table for the sake of space.

Why would one even want to activate p73 for cancer research?

What are the main conclusions for chapter p73 activation?

252: unclear: “several of these cell lines

Chapter 6: I feel it is unclear what is the consequence of virus induced loss of function and why does it matter in the sense of your article and cancer? Please explain more clearly at the beginning of the chapter.

Line 282: Please remove “ΔN”.

283: It is not native, but negative. Please check the whole text.

Chapter 8 doesn’t seem to be Conclusion as it was named. Please revisit. Why 289-309 are not part of chapter 7? Maybe the first sentence in Chapter 8 can be similarly said at the beginning of chapter 6 and 7 as well to make it clearer.

It seems that Conclusion actually spreads through lines 309-312.

Line 312-320 is not a part of conclusion, please move elsewhere.

Overall, the article provides many important information, but some clarification and organisation is needed.

Thank you.

Round 2

Reviewer 2 Report

Dear Authors, thank you for your responses to the suggestions and comments.

Here are further suggestions:

Line 23  "and full-length p53 family members all have a tumor suppressor function." This sentence is not correct. Please check the following papers and make this statement a bit softer. One may not simply claim that all full lenght isoforms are tumor suppresors.

https://doi.org/10.3389/fcell.2021.737735

https://www.mdpi.com/2072-6694/13/12/2885

https://www.mdpi.com/1422-0067/20/24/6257

https://doi.org/10.1007/s00018-017-2666-y

Line 34 " Full length p53 family members are tumor suppressor genes" This sentence is not correct.  Please check the following papers and make this statement a bit softer. One may not simply claim that all full lenght isoforms are tumor suppresors.

https://doi.org/10.3389/fcell.2021.737735

https://www.mdpi.com/2072-6694/13/12/2885

https://www.mdpi.com/1422-0067/20/24/6257

https://doi.org/10.1007/s00018-017-2666-y

Line 27-29 "Because p73 is widely 27 expressed in organs compared to p63, which has limited expression, there are many more p73 activators than p63 activators" This doesnt have to always be the case. I don't know why is it important to discuss why there are more p63 activators. And if there is a need for this, maybe make this sentence a bit softer.

Line 59 Please revise "mutant" 

Line 59-60: "Each mutant contains a >2% p53 mutation rate in breast cancer at codons amino acid 175, 179, 195, 196, 213, 217, 225, 229, 248, 249, 254, 60 273, 278, 281, and 298 [20]." Why did the authors choose this paper and breast cancer for this part of the paper, as I'm sure there are many other papers dealing with p53 mutation rate in other models too. Why are the aminoacid positions important here?

Line 66: There is extra "syndrome"

Line 71: First "mutation" is extra

Line 78:  Please check again the article under refenrence 29 because "The major isoform of p63 is dominant-negative  Np63α; TAp63α is only expressed in specific organs" is incorrect and please, correct your text accordingly. Here is the original wording from the paper you cited: "In all tissues except skeletal muscle, p63α was the predominant (67–97%) isoform expressed" and "∆Np63 was the predominant isoform (80–100%)"

Line 98: Reference under 29 doesnt actually destinguish C-terminal and N-terminal isoforms at the same time. Please check your statement.

Line 98: Reference 37 writes: "TAp73 isoform is abundantly expressed in cancer cell lines compared to the dominant negative ΔNp73 isoform"and "Surprisingly, we found consistently higher expression of TAp73 isoforms in the selected cell lines," Please check your statement.

Line 98-99: "So more p73 activators develop compared to p63 activators owing to the limited expression of TAp63α." This is not logical and it doesnt have to always be the case. 

Line 117: Please explain is this something you have calculated by yourself or is it in the paper "theoretical rate of nonsense mutants 116 (3/64=4.6%)."

Line 140: The text would flow better if the information that "missense mutations contains both loss of function and gain of function" was added in the text.

Line 145: Please revise: "mutants p53" 

Line 198: p63 isoforms according to this reference are not widely expressed in tissues, since "The majority of tissues expressed low-levels of both TP73 and TP63"

Line 214: Please revise : "Coding sequence (CDS), is the portion of a gene's DNA that codes for protein." to  "CDS, coding sequence".

Line 217: p73 isoforms according to the reference 29 are not widely expressed in tissues, since "The majority of tissues expressed low-levels of both TP73 and TP63" Reference 37 is refering mostly to cancer cell lines, not on organs, so please specify this.

Line 221: "NSC59984 can induce mutant p53 protein degradation TO"  is not correct. Please revise to "NSC59984 can induce mutant p53 protein degradation AND"

Line 259-277:  It is unclear what is the meaning of this part. Maybe a title would help, or a brief introductory sentence. Was the purpose of this part to discuss the influence of interactions between p53 family members on p63 or p73 activators and the importance of combination treatment strategy. If, so, please write a sentence. 

Line 263 is almost exactly the same as line 265-266.

Line 268: Please revise the expression "mutant p53 aggregative cells"? Are cells "aggregative"?

Line 301: "Two major oncogenic type" Please be aware that p53 isoforms are not yet proven to be exactly oncogenic nor tumor suppressive. See references suggested above. Please make this statement a bit softer. 

Line 310: The p53 isoforms stated here: "One ∆N isoform of p53, Δ133p53β, but not Δ133p53α" are not ∆N isoforms, but isoforms of both ends, so please remove ∆N or remove altogether: "One ∆N isoform of p53,".

Line 342: Would transtering the lines 342-350 before 337 sound better?

Thank you

Kind regards.

Round 3

Reviewer 2 Report

Dear Authors, 

thank you for your response, and accepting my suggestions.

I still have some concern regarding lines:

23 and 35: Please can you consider avoiding the word "all" simply to make the statement around tumor suppressor role of ALL p53 TA isoforms a bit softer. This is very important. If you check the references under:

https://doi.org/10.3389/fcell.2021.737735
https://www.mdpi.com/2072-6694/13/12/2885
https://www.mdpi.com/1422-0067/20/24/6257
https://doi.org/10.1007/s00018-017-2666-y

you may see that even TAp63 can act as oncogene, but I agree that generally TA isoforms or full lenght isoforms are considered tumor suppressors. Maybe your former "full lenght" expression is a better choice, since TA isoforms can have different functions depending on the C-terminus.

So, maybe writing: "The members of the p53 family comprise p53, p63 and p73, and full-length isoforms of p53 family have a tumor suppressor function."

Line 68: LFS already contains "syndrome" in the abbrevitation, so there is no need to write LFS syndrome, just LFS is enough.

Line 80: The text: "The major isoform of p63 is dominant-negative â–³Np63α expressed in the skin, esophageal mucosa, vagina, and prostate; TAp63α is only expressed in specific organs testis and skeletal muscle"

is not correct.

The method from this paper wasnt actually able to distinguish both N terminal and C terminal isoforms at the same time.

And it states that out of C-terminal isoforms, alpha isoforms are predominant: " In all tissues except skeletal muscle, p63α was the predominant (67–97%) isoform expressed".

And further when N-terminal isoforms (delta N and TA) were looked into they state that "ΔNp63 was the predominant isoform (80–100%) in most tissues from the p73-High/p63-High group".

Please revise your text accordingly. Suggestion: "The major C-terminal p63 isoforms are p63α, while dominant-negative ΔNp63 was the predominant N-terminal isoform in most tissues from the p73-High/p63-High group."

Thank you.

Kind regards and I hope this review finds you well. 
